



# Towards a global impact-based forecasting model for tropical cyclones

Mersedeh Kooshki Forooshani[1], Marc van den Homberg[2], Kyriaki Kalimeri[1], Andreas Kaltenbrunner[1,4], Yelena Mejova[1], Leonardo Milano[3], Pauline Ndirangu[3], Daniela Paolotti[1], Aklilu Teklesadik[2], and Monica L. Turner[3]

[1]ISI Foundation, Turin, Italy
[2]510, an Initiative of the Netherlands Red Cross, The Hague, Netherlands
[3]UN OCHA Centre for Humanitarian Data, The Hague, Netherlands
[4]Internet Interdisciplinary Institute, Universitat Oberta de Catalunya, Barcelona, Spain

**Correspondence:** Marc van den Homberg, mvandenhomberg@redcross.nl; Andreas Kaltenbrunner, kaltenbrunner@gmail.com

**Abstract.** Tropical cyclones (TCs) produce strong winds and heavy rains accompanied by consecutive events such as landslides and storm surges, resulting in losses of lives and livelihoods particularly in regions where socioeconomic vulnerability is high. To proactively mitigate the impacts of TCs, humanitarian actors implement anticipatory action. In this work, we build upon such an existing anticipatory action for the Philippines, which uses an impact-based forecasting model for housing damage based on

XGBoost to release funding and trigger early action. We improve it in three ways. First, we perform a correlation and selection analysis, to understand if Philippines-specific features can be left out or replaced with features from open global data sources. Secondly, we transform the target variable (percentage of completely damaged houses) and not yet grid-based global features to a 0.1 degrees grid resolution by de-aggregation using Google Building Footprint data. Thirdly, we evaluate XGBoost regression models using different combinations of global and local features at both grid and municipality spatial level. We introduce a

two-stage model to first predict if the damage is above 10% and then use a regression model trained on either all or on only high damage data. All experiments use data from 39 typhoons that impacted the Philippines between 2006-2020. Due to the scarcity and skewness of the training data, specific attention is paid to data stratification, sampling and validation techniques. We demonstrate that employing only the global features does not significantly influence model performance. Despite excluding local data on physical vulnerability and storm surge susceptibility, the two-stage model improves upon the municipality-based

model with local features. When applied for anticipatory action our two-stage model would show a higher True Positive rate, a lower False Negative rate and furthermore an improved False Positive rate, implying that fewer resources would be wasted in anticipatory action. We conclude that relying on globally available data sources and working at grid level holds the potential to render a machine learning-based impact model generalisable and transferable to locations outside of the Philippines impacted by TCs. Also, a grid-based model increases the resolution of the predictions, which may allow for a more targeted

implementation of anticipatory action. However, it should be noted that an impact-based forecasting model can only be as good as the forecast skill of the TC forecast that goes into it. Future research will focus on replicating to and testing the approach in other TC-prone countries. Ultimately, a transferable model will facilitate the scaling up of anticipatory action for TCs.



## 1    Introduction

The emission of greenhouse gases due to human activity in the past decades has had a significant effect on global climate
variability, and the resulting climate change will likely increase conditions that shape extreme events such as Tropical Cyclones
(TCs, Van Aalst, 2006). TCs are massive storms that form over warm tropical oceans and cause extreme rainfall (Navarro and
Merino, 2022), leading also to consecutive events such as landslides (Jones et al., 2023), storm surges (Bloemendaal et al.,
2019), and floods (Eilander et al., 2022). Over 20 million people have been affected by TCs, and almost 30 billion US$ in
damages have been reported yearly in the last two decades (Geiger et al., 2018). For 2022, the Emergency Events Database
(EM-DAT, 2022) reports that 36.9 million people were affected by storms, and these storms were responsible for 90.2 billion
US$ of economic loss. TCs occur in many parts of the world but mainly in North America, East Asia, and the Caribbean–Central
American region (Gettelman et al., 2018; Mendelsohn et al., 2012). Significantly, the population and infrastructure close to the
coast (Rogers et al., 2019) get impacted by TCs. Developing countries are disproportionately affected, as their population is and
will -with climate change- be more exposed (Bloemendaal et al., 2022) and more vulnerable to TCs due to their socioeconomic
conditions (Hallegatte et al., 2016).

Until recently, humanitarian action has been primarily reactive, only initiating a response after a disaster. However, over
the past decade, the increased amount of data availability and improved weather forecasting capability has enabled humani-
tarian actors to implement anticipatory action (AA), focusing on reducing the impacts of a hazard before it occurs (van den
Homberg et al., 2020). AA plays a crucial role in enabling humanitarian organizations to mitigate the impact of various shocks
proactively, and recent evidence suggests that AA is more dignified, swift, and cost-effective than humanitarian response
(Chaves-Gonzalez et al., 2022).

AA triggers are built using hazard- or impact-based forecasts (Harrison et al., 2022). If the forecast exceeds a predetermined
threshold (with a certain probability), early actions are implemented to save lives and protect property and livelihoods (Yon-
son et al., 2018). Around the world, many governmental and humanitarian actors are working hand in hand to develop AA
mechanisms (Anticipation Hub, 2022). The Red Cross National Societies of seven countries have implemented AA for tropical
cyclones with a large group of in-country stakeholders, i.e., Bangladesh, Mozambique, the Philippines, Costa Rica, Guatemala,
Honduras, and Madagascar. Similarly, UN OCHA has piloted AA in multiple countries for several hazards, particularly tropical
cyclones, together with the Philippine Red Cross (ReliefWeb, 2022).

Among the many countries at risk of tropical cyclones, the World Risk Index 2022 [1] put the Philippines at the number one
spot for the most-disaster-prone country in the world (Atwii et al., 2022). The Philippines is recognized as a global "hot spot"
for natural hazards and endures a higher frequency of disasters due to earthquakes, typhoons, floods, and landslides than any
other country, with an average of eight or nine disasters annually (Santos, 2021). After reviewing TC data spanning from 1951
to 2013 in the Philippines found that the Philippine Area experiences an average of nearly 20 TCs every year (Cinco et al.,
2016).

---

[1]www.WorldRiskReport.org





In 2016, the 510 initiative of the Netherlands Red Cross started working with the Philippine and German Red Cross to develop a model to predict the humanitarian impact of typhoons. Initially, the emphasis was on understanding the needs of the humanitarian decision-makers and collecting and collating data on several features and target values of the model through desk research and in-country visits of key stakeholders (Van Lint et al., 2016). Since the first model in 2016, this model, which, for simplicity, we will refer to in the remainder of the paper as the 510 model, has undergone many iterations to improve its

performance further. In 2019, the 510 model (Teklesadik et al., 2023; Teklesadik and van den Homberg, 2022) was approved as the trigger model for the Early Action Protocol for typhoons[2] and in 2021 as the trigger model for the UN OCHA AA pilot. This approach combines historical impact and vulnerability data with typhoon tracks and weather forecasts to generate early estimates of the expected damage of a typhoon before landfall. The model was specifically built for the Philippines,

In this study, we pursue two goals. First, due to the global prevalence of TCs and their disproportional impact on developing

countries, we aim to extend the 510 model to other geographical contexts to create a globally applicable impact model for TCs. For this purpose, we select features that we can use for different geographical contexts (i.e., countries) because the data for these features can be selected from open-access global databases. Secondly, we seek to ensure the model's performance is not deteriorated in this process and, if possible, improved.

The 510 model is a probabilistic typhoon impact prediction model whose spatial configuration is vector-based where model

inputs are aggregated per municipality (Teklesadik et al., 2023; Teklesadik and van den Homberg, 2022). This approach was chosen due to the usage of localized datasets collected at the municipality level. However, there are two reasons why a grid-based model configuration should be a better approach to test the hypothesis of porting the typhoon model to other contexts. Firstly, because open datasets, for example, for hazard, exposure, and vulnerability, are often grid-based, and secondly, because such models become independent of the specific geographic resolution of administrative regions in a given territory. To test

this hypothesis, we assess the performance of a variant of the 510 model still in the context of the Philippines but only using globally available variables. We then implement a model with grid-based spatial configuration using only the globally available features. To compare the performance of this grid-based model with the 510 model, we transform its prediction results back to the municipality level. Finally, to achieve better performance of the grid-based model, we include additional globally available features that were not used in the 510 model and build a novel two-step prediction model. We illustrate the capacity of this new

approach concerning correctly predicting damage levels above a given threshold, which would trigger early actions. Our results allow us to conjecture about the feasibility of generalizing our particular grid-based model to other countries and reducing the impact of humanitarian crises, with the ultimate goal of saving lives and protecting livelihoods from disasters due to TCs.

## 2   Related Literature

Disasters manifest in various regions globally, driven by a confluence of hazard occurrence, exposure levels, and the vulnerabil-

ity of human populations and valuable assets. Historically, National Meteorological and Hydrological Services (NMHSs) only focused on furnishing weather-related information and warnings based on meteorological factors such as wind speeds, rainfall,

---

[2]https://reliefweb.int/report/philippines/philippines-typhoon-early-action-protocol-summary-november-2019



and hazard location and timing. Nevertheless, in the past decade, NMHSs and their collaborating agencies have made substantial efforts to enhance their comprehension of the potential repercussions of severe hazards. Achieving this goal necessitates robust partnerships with collaborating agencies and extensive research into impact-based forecasting models, incorporating exposure and vulnerability data.


Early studies assessed the impacts of floods and TCs in different aspects. In one of these first studies by Vickery et al. (2006), a Hurricane Model named HAZUS-MH was developed to predict the building damage caused by hurricanes in the USA. This model has been validated using damage data collected during post-storm damage surveys and insurance loss. Later, in a study by Liu et al. (2009), historical data of typhoon disasters in China was used to prevent and mitigate the life and property losses due to these phenomena in New Orleans and Shanghai. The study found a stronger correlation between wind speed and water level than other variables.


Another early study about coastal flood risks was done by Boettle et al. (2011) and was based on estimating typical damages caused by storm surges. This study determined that although the damage depends on various factors, such as flow velocity, flood duration, etc., the correlation between flood occurrences and the average damages is typically explained using a stage-damage function, which employs the maximum water level as the only damage influencing factor.


In a study by Wagenaar et al. (2018), a flood damage model was proposed, using Random Forests and Bayesian Networks to estimate the residential damage based on water depth and average building value. The study leveraged data from Germany and the Netherlands to cross-validate the model performances. Alternatively, Kim et al. (2019) used regression models to determine whether typhoon damages are correlated with wind speed, rainfall, and the number of cutting slopes and then assess the impact of built environment vulnerability on financial loss using typhoon data. More recently, a hybrid model using Convolutional Neural Networks (CNN) and Long Short-Term Memory (LSTM) was introduced by Chen et al. (2019) as a predictive model of Western North Pacific typhoon formation and intensity with an emphasis on the various spatial and temporal features of typhoons.


In 2020, a statistical prediction model was proposed by Kim et al. (2020) for China. Its data included daily rainfall data for 55 typhoons between 1961 and 2017 from 537 meteorological stations in China. The model was based on the principle of track similarity and used different methods, such as fuzzy C-means clustering and intensity correction. This model aimed to improve the typhoon-induced accumulated rainfall forecasts over China. In another recent study, a group of researchers (Hou et al., 2020) built a hybrid model to predict the damage probability of transmission lines under each wind field for a particular typhoon named Mangkhut'2018 in China. It used the Monte Carlo method to simulate the random wind field to improve the prediction with Random Forests.



For the specific context of the Philippines as one of the most climate disaster-prone countries, some research has been done in the past few years. Recently, Wagenaar et al. (2021) made notable contributions by proposing models based on the Random Forest and Artificial Neural Networks. They used data from 12 typhoons in the Philippines at the municipality level to explain the relationships between damages and the variables that can explain damage, such as water depth or wind speed. The Red Cross collected this dataset and includes 40 variables from which damage is predicted. In another recent study, Lambert et al. (2022) utilized the following machine learning (ML) algorithms: Random Forest (RF), k-nearest Neighbors (KNN), and generalized




linear models (GLM) to predict the damage caused by urban forest storms. They reported that GLM and RF models gave overall unbiased damage predictions across all methods and rarity levels, while KNN consistently under-predicted damage. A vulnerability risk model for the Philippines was put forward by Baldwin et al. (2023), in which they assess the vulnerability using the wind field data and total asset value and determine the expected asset loss. In a study, Walsh (2020) proposed a traditional expanded risk assessment using asset losses as the primary metric to measure the severity of a disaster.

Finally, the already mentioned 510 model (Teklesadik et al., 2023) was developed recently by the Netherlands Red Cross as a vector-based or municipality-based prediction model to estimate the damage to houses caused by TCs in the Philippines. It used data from 39 typhoons with mild to severe damage impact, and the independent variables included 36 features related to hazard characteristics and vulnerability data, all at the municipality level. In this study, we aim to expand the applicability of this model to other contexts by constraining its feature set to internationally-available data while improving its performance in the Philippines setting.

## 3  Data & Methodology

This study is based on data and features employed by Teklesadik and van den Homberg (2022) in the 510 model (See also (Teklesadik et al., 2023)) to train a typhoon impact-based forecasting model. It includes data from 39 typhoons that impacted the Philippines between 2006 and 2020 collected from various organizations and resources and at different spatial resolutions. For example, data on damaged houses is collected at the individual housing level but only available with open access at an aggregated municipality level (i.e., admin level 3 in the Philippines). We extend this model by using additional data (available on a global scale) and improve the prediction and evaluation methods. Table 1 shows the features used in previous work (the 510 model), the ones added in this study, and their descriptions. Note that the model used by Teklesadik et al. (2023) operated at the municipality level while the models we developed use data in a 0.1-degree grid format[3] whose area is smaller than the average size of a municipality. Although most of the features in our developed grid-based models are directly available at the 0.1-degree resolution, some are transformed to grid resolution after being obtained from their sources at the municipality level. Below, we describe the features, the target variable, and the transformation process (when applicable) in more detail.

---

[3] Approx. 11x11 km$^2$. This spatial resolution could be increased for other contexts.



**Table 1.** Description of features employed by the different models. Features 1-36 are used as municipality and grid-level resolution while features 37-42 are only used at the grid-level resolution. For better comprehension, the labels of some features may differ with respect to their original labels in the input data.

| no | Feature Label | Description | Local | Global | Global+ |
|---|---|---|---|---|---|
| 1 | HAZ_rainfall_Total | Total volume of rain during a typhoon event | ✓ | x | x |
| 2 | HAZ_rainfall_max_6h | Maximum rainfall within a 6 hour period (mm) | ✓ | ✓ | ✓ |
| 3 | HAZ_rainfall_max_24h | Maximum rainfall within a 24 hour period (mm) | ✓ | ✓ | ✓ |
| 4 | HAZ_v_max | Max. 1-min. sustained windspeed, based on Windfield (m/s) | ✓ | ✓ | ✓ |
| 5 | HAZ_v_max_3 | Max. 1-min. sustained windspeed cubed, based on Windfield (m/s) | ✓ | x | x |
| 6 | HAZ_dis_track_min | Minimum distance between typhoon track and municipality | ✓ | ✓ | ✓ |
| 7 | HAZ_SEC_landslide_per | % of houses (OSM footprint) in landslide risk zones (red, yellow, orange) | ✓ | x | x |
| 8 | HAZ_SEC_stormsurge_per | % of houses (OSM footprint) in storm surge risk zones (red, yellow, orange) | ✓ | x | x |
| 9 | HAZ_SEC_Bu_p_inSSA | Fraction of municipality coloured blue in storm surge risk map | x | x | x |
| 10 | HAZ_SEC_Bu_p_LS | Fraction of municipality coloured blue in landslide risk map | x | x | x |
| 11 | HAZ_SEC_Red_per_LSbldg | Fraction of municipality coloured red in land slide risk map | ✓ | x | x |
| 12 | HAZ_SEC_Or_per_LSblg | Fraction of municipality coloured orange in land slide risk map | ✓ | x | x |
| 13 | HAZ_SEC_Yel_per_LSSAb | Fraction of municipality coloured yellow in storm surge risk map | ✓ | x | x |
| 14 | HAZ_SEC_RED_per_SSAbldg | Fraction of municipality coloured red in storm surge risk map | x | x | x |
| 15 | HAZ_SEC_OR_per_SSAbldg | Fraction of municipality coloured orange in storm surge risk map | ✓ | x | x |
| 16 | HAZ_SEC_Yellow_per_LSbl | Fraction of municipality coloured yellow in storm surge risk map | ✓ | x | x |
| 17 | TOP_mean_slope | Slope mean | ✓ | ✓ | ✓ |
| 18 | TOP_mean_elevation_m | Elevation mean | ✓ | ✓ | ✓ |
| 19 | TOP_ruggedness_stdev | Ruggedness standard devation | ✓ | ✓ | ✓ |
| 20 | TOP_mean_ruggedness | Ruggedness mean | x | ✓ | ✓ |
| 21 | TOP_slope_stdev | Slope standard devation | x | ✓ | ✓ |
| 22 | TOP_with_coast | Boolean: coast or no coast | ✓ | ✓ | ✓ |
| 23 | TOP_coast_length | Length of coast | ✓ | ✓ | ✓ |
| 24 | VUL_poverty_perc | Percentage of people in poverty | ✓ | x | x |
| 25 | VUL_Housing_Units | Total number of housing units | ✓ | ✓ | ✓ |
| 26 | VUL_StrongRoof_StrongWall | Number of houses with a strong roof and strong walls | ✓ | x | x |
| 27 | VUL_StrongRoof_LightWall | Number of houses with a strong roof and light walls | ✓ | x | x |
| 28 | VUL_StrongRoof_SalvageWall | Number of houses with a strong roof and salvaged walls | ✓ | x | x |
| 29 | VUL_LightRoof_StrongWall | Number of houses with a light roof and strong walls | ✓ | x | x |
| 30 | VUL_LightRoof_LightWall | Number of houses with a light roof and light walls | ✓ | x | x |
| 31 | VUL_LightRoof_SalvageWall | Number of houses with a light roof and salvaged walls | ✓ | x | x |
| 32 | VUL_SalvagedRoof_StrongWall | Number of houses with a salvaged roof and strong walls | ✓ | x | x |
| 33 | VUL_SalvagedRoof_LightWall | Number of houses with a salvaged roof and light walls | ✓ | x | x |
| 34 | VUL_SalvagedRoof_SalvageWall | Number of houses with a salvaged roof and salvaged walls | ✓ | x | x |
| 35 | VUL_vulnerable_groups | Vulnerable groups from DSWD National Household Targeting Office | ✓ | x | x |
| 36 | VUL_pantawid_pamilya_beneficiary | Number of Pantawid Pamilya beneficiary households | ✓ | x | x |
| 37 | VUL_relative_wealth_index (rwi) | Relative standard of living within countries | x | x | ✓ |
| 38 | total_pop | The total population | x | x | ✓ |
| 39 | urban | Proportion of urban areas | x | x | ✓ |
| 40 | rural | Proportion of rural areas | x | x | ✓ |
| 41 | water | Proportion of areas classified as water | x | x | ✓ |
| 42 | Percent_houses_damaged_5years | Percentage of damaged houses in last 5 years | x | x | ✓ |





## 3.1 Target Variable

The target variable of the models analyzed in this study is the percentage of fully damaged houses at the municipality level. The Department of Social Welfare and Development collects this data at the individual house level, assigning a partially or fully damaged label to each house (Office, 2019). A house is a dwelling or structure used for human habitation, especially by a family or small group. Based on this damage label, people are eligible for emergency shelter assistance. The data is only open access at the aggregated level of the municipality. We use the percentage of houses fully damaged and unfit for habitation or without any remaining structural features. This damage variable can vary in the range between 0 and 100%.

Housing damage is improbable to occur under low rainfall and wind speed conditions. Hence, missing damage percentage values in the original data of the municipality-based 510 model were replaced with zero for records with low wind speed (below 25m/s) and rainfall (below 50mm). In contrast, the remaining entries with missing damage values that did not fulfill these conditions have been removed from the dataset. It should be noted that these thresholds are defined based on long-term average observation of climate data.

To improve the resolution of the original 510 model, we transform the original values of damage data from the municipality level to the grid format. To do so, we use the number of buildings from Google Building Footprint data[4] (See Figure ?? for a visualization of this data) to compute transformation weights. We also checked Microsoft Building Footprint and OSM building delineation data, but these datasets were incomplete for the Philippines. Specifically, for a given municipality, we count the number of buildings in each grid cell with a geographical intersection with the municipality and then normalize by the number of buildings in a municipality. In this way, we give more weight to the grid cells in which a municipality has more buildings.

For the evaluation, which is done at the municipality level, we perform the opposite transformation: We normalize by the number of buildings in a grid cell to get the back-transformation weights from grid cells to municipalities. Note that these transformations are not bijections, in the sense that, for example, in a grid cell where only one of its intersecting municipalities has damage larger than 0, during re-aggregation, this damage score will be distributed among the neighboring municipalities intersecting with the grid cell.

Figure 2. shows the distributions of both the original damage variable used by the 510 model (blue) and the re-aggregated (red) damage variables for all typhoons. Note that, after the transformation back from the grid, we gain more data points in the lowest damage area due to the re-distribution of damage data to neighboring municipalities – a property that affects evaluation metrics (as will be discussed in the Results section). Furthermore, as the distribution is heavily skewed (see left figure), we bin the data in the following intervals: [0, 0.00009], (0.00009, 1], (1, 10], (10, 50], (50,100]. We use these bins for stratification during model training and to compute performance metrics for each bin separately.

---

[4]https://sites.research.google/open-buildings/#download



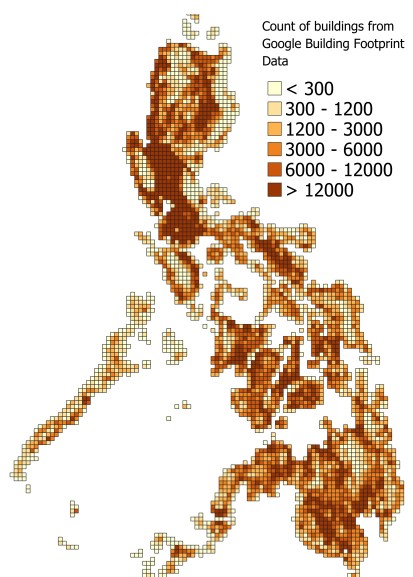

**Figure 1.** The number of the building centroids, from Google Open Buildings dataset, aggregated to a 0.1-degree grid. The lighter shades of brown indicate grids with fewer buildings while the darker shades point to grids with a higher count of buildings.

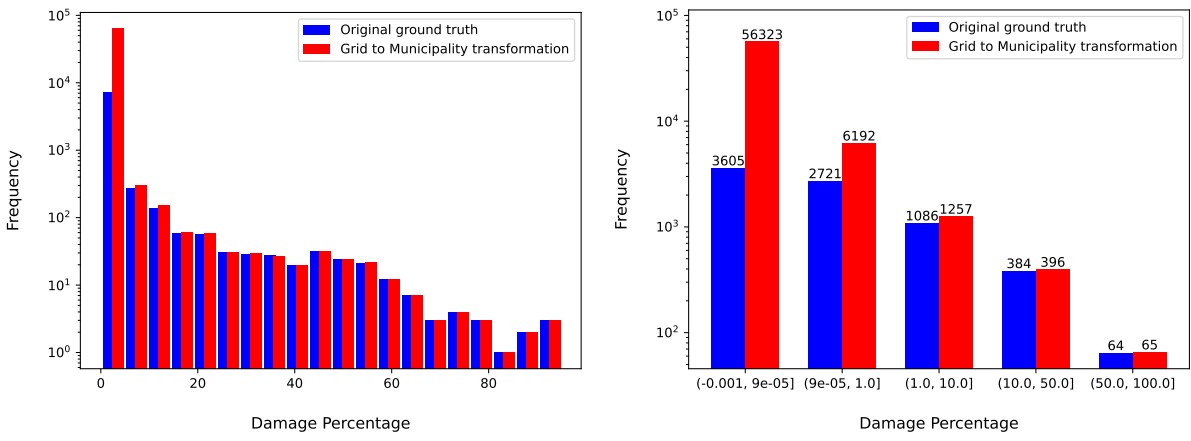

**Figure 2.** Distribution of damage % at the municipality level. Right shows the stratification by unequally-sized bins used for training.

## 3.2 Original Features of the Municipality-based Model

In this section, we describe those features used in the original 510 model of Teklesadik et al. (2023). As input data for our models, we use both municipality and grid-based versions of these features. We will indicate this by adding M (for municipality) or G (for grid) as prefixes to the model names. Rows 1 to 36 of Table 1 briefly describe those features, which we extend below. Furthermore, the table also indicates different groups of features in the three rightmost columns, which we label Local, Global,




and Global+. They are used in the suffixes of the model names to indicate which subset of features the model uses. We note
that the source data for those features used in the Local, Global, and Global+ models are the same, so the Local model has
features for which the source data was at the grid level.

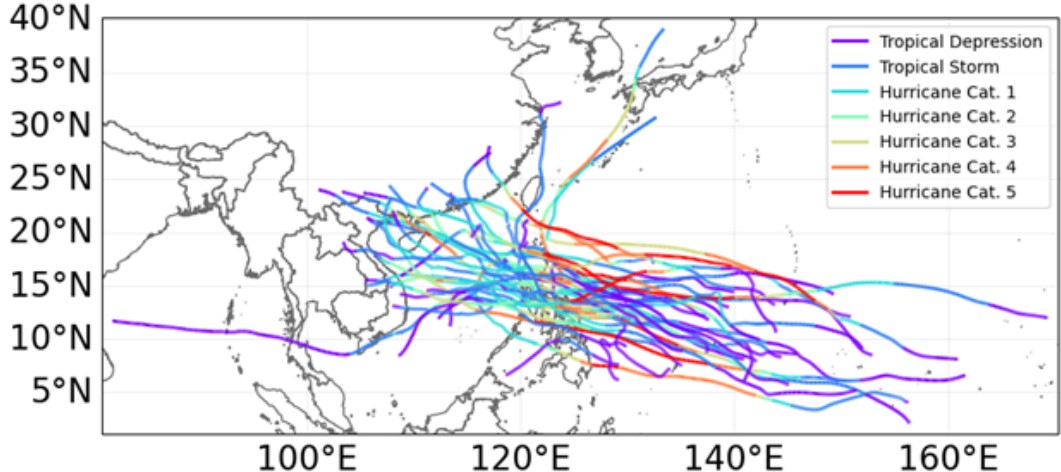

**Figure 3.** TC track data for the Philippines Area of Responsibility with the colours indicating the tropical cyclone intensity. The categorize
from 1 to 5 refer to the weakest and the strongest, respectively. The naming protocols for TCs vary by region. Typhoon is mostly used in the
western North Pacific region which includes the Philippines. However, IBTrACS data use the hurricane label, regardless of the TC region.

1. Features 1-6, with the prefix HAZ, are created from historical typhoon and weather metadata. The source of these features
   was at the grid level except for the Typhoon track data (6. HAZ_dis_track_min), which was directly obtained at the mu-
   nicipality level. Rainfall is obtained from the NASA Precipitation Processing System (PPS), which aggregates weather
data from the Global Precipitation Measurement (GPM) project[5]. In particular, we use the maximum of the correspond-
   ing 6 or 24-hour rolling average windows of the 30-minute GPM rainfall data. At the same time, the HAZ_rainfall_Total
   is the sum over all rainfall in a $\pm$ three-day period of typhoon landfall.

   The typhoon track data was collected from the International Best Track Archive for Climate Stewardship (IBTrACS)[6].
   See Figure 3 for a visualization of the tracks of the typhoons used in this study. We see a great variety of the spatial
distribution of typhoons as they impact different regions of the Philippines.

   The maximum wind speed per municipality was estimated by generating wind fields using Climada[7] and the default
   Holland 1980 model with B parameter from Holland 2008, which determines the shape of the wind profile (Holland,
   1980, 2008).

---

[5]https://arthurhou.pps.eosdis.nasa.gov/

[6]https://www.ncei.noaa.gov/products/international-best-track-archive

[7]https://wcr.ethz.ch/research/climada.html



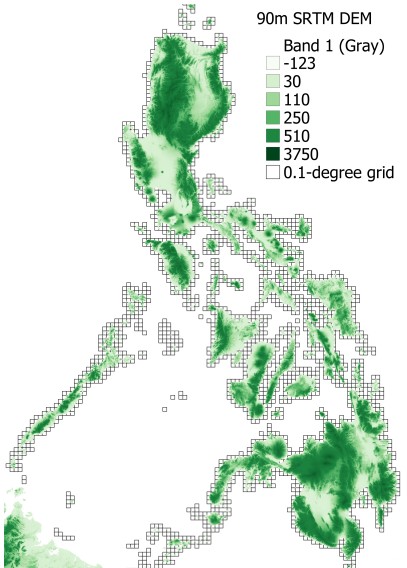

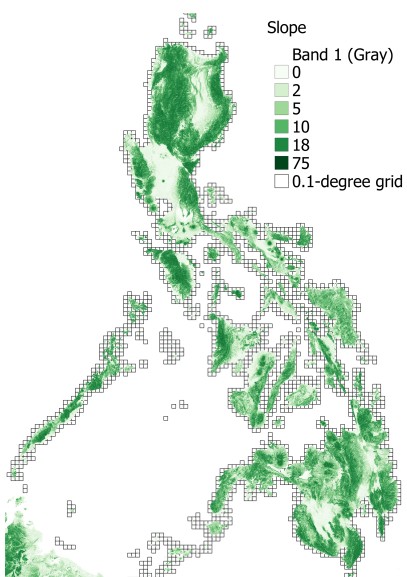

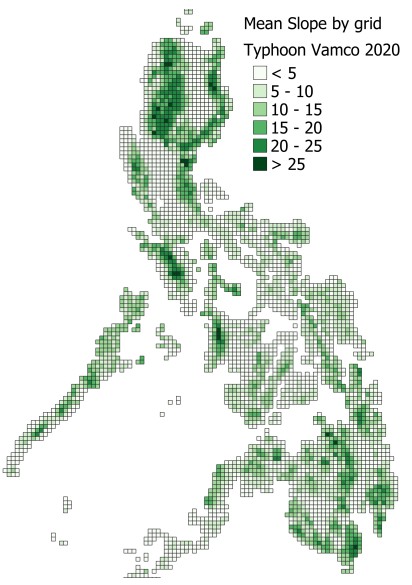

**Figure 4.** Elevation data from the SRTM database. It shows the elevation at 90m spatial resolution. Light green indicates lower elevation while dark green shows high elevation.

**Figure 5.** Slope, computed from the SRTM database. The slope is still shown at a 90m spatial resolution with the shades of green indicating the steepness of the slope. Light green shows a flatter slope while dark green shows a steep slope.

**Figure 6.** Mean slope by grid. The 90m slope is averaged for each 0.1-degree grid.

2. Features 7-16 with the prefix HAZ_SEC are data for landslide and storm surge vulnerable areas. The source of these features was at the grid level. Hazard maps were taken from the National Operational Assessment of Hazards (NOAH)[8], and the fraction of each risk level (set by color) intersecting with each municipality was used to create the features. Unfortunately, these maps are no longer available online.

3. Features 17-23 with the prefix TOP are the topography data, including features related to the slope, terrain ruggedness, elevation, and coastline length. The source data of these features was available at the grid level except for the coastline features (22.TOP_with_coast and 23.TOP_coast_length). These were collected via GIS analysis combined with the Common Operational Datasets for the Philippines [9]. All but coastline length were generated using 90m SRTM DEM (Shuttle Radar Topography Mission Digital Elevation Model) from CGIAR CSI[10]. Figure 4 shows the corresponding elevation data and Figure 5 the slope data derived from the elevation data, and finally in Figure 6 we show the mean slopes

---

[8]https://noah.up.edu.ph/

[9]https://cod.unocha.org/

[10]https://srtm.csi.cgiar.org/





aggregated by grid cells. The coastline length was computed from the Common Operational Dataset Administrative

Boundaries (COD ABs)[11] for the Philippines.

4. Feature 24 with the prefix VUL represents the percentage of people in poverty. It was generated from an analysis of the 2012 census[12].

5. Features 25-36 were all synthesized from Philippines Pre-Disaster Indicators datasets on the Humanitarian Data Exchange[13]. Feature 25 is simply the number of houses per municipality. For this feature we transformed the original

values from the municipality level to the grid level by using building data as explained in section 3.1. Features 26-34 denote the composition of the housing construction materials. Feature 35 represents the number of vulnerable groups by city/municipality from DSWD National Household Targeting Office, while feature 36 is based on the number of Pantawid Pamilya[14] beneficiary households. For all these features starting with the prefix "VUL", the original source was at the municipality level.

**3.3    Additional grid-level only Features**

In addition to the features described above, we also add features from globally available datasets that were available in grid level (contributing to the set of features we dub Global+), but sometimes in a different resolution than our 0.1 grid. We describe them below and explain the corresponding transformation, if needed.

6. Feature 37 with the prefix VUL corresponds to the Relative Wealth Index (rwi)[15] which was mean-aggregated for each

grid cell. It originally came as point data and the corresponding grid value was derived from points contained wholly within a grid cell. Since there exist missing values in some grid cells of this feature, we estimated the average of available values over all grid cells and then replaced null values with this average to diminish the impact of missing data and preserve as much as possible the data integrity.

7. Feature 38, the population data[16] is available as a 100m resolution raster and is aggregated to each grid cell.

8. Features 39-41 represent the proportion of urban, rural and water areas aggregated similarly to the population data. They are based on the 2025 epoch of the Degree of Urbanisation dataset[17] from the GHSL (Global Human Settlement Layer) which classifies settlement typologies and has a 1km resolution raster. The GHSL dataset provides complete information on human settlements built on satellite imagery and other geospatial data. To calculate the proportion of urban areas for one of our 0.1-degree grid cells, we take the fraction of it which has values of 21 or greater in the Degree of Urbanisation

---

[11]https://cod.unocha.org/

[12]Unfortunately the original dataset and analysis are no longer publicly available

[13]https://data.humdata.org/dataset/philippines-pre-disaster-indicators

[14]https://pantawid.dswd.gov.ph/

[15]https://data.humdata.org/dataset/relative-wealth-index

[16]https://ghsl.jrc.ec.europa.eu/download.php?ds=pop

[17]https://ghsl.jrc.ec.europa.eu/download.php?ds=smod





dataset. Similarly, the proportion of rural areas is the percentage that has values between 11 and 13, and the proportion

of water the percentage values of 10. The sum of these three features add up to 1.

9. Feature 42, finally is the percentage of damaged houses in the five years prior to a typhoon event, calculated as the

average of the target variable in the five years prior to the disaster event. This value is 0 in the absence of any prior data.

## 3.4   Feature selection

Selecting the most important features and their relevance in a dataset aids in effectively applying ML algorithms in real-world

scenarios. Therefore, in this study, we use correlation among features to select features that reduce multicollinearity. See Figure 7 for a visualization of the feature correlations in the municipality dataset. Using this information, we remove features number 9 (HAZ_SEC_Bu_p_inSSA), 10 (HAZ_SEC_Bu_p_LS), 14 (HAZ_SEC_OR_per_SSAbldg), 20 (TOP_mean_ruggedness),

and 21 (TOP_slope_stdev) in Table 1 from the input municipality dataset since they correlate more than 0.99 with other fea-

tures.

## 3.5   Predictive Models

Our models are trained using XGBoost (both for regression and classification), a popular tree-based ensemble-learning method, which was also used in the 510 model, which our analysis extends. Additionally, we compared our models' performance with a naive baseline (based on the average of training data) and Linear Regression and Random Forest. We omit results for the

latter two models from the analysis below as they perform slightly more poorly. We describe all the models we used and the

corresponding set of features in Table 2.

**Table 2.** Table of Models Description.

| Model Name | # Features | Model Description | Feature Set (see Table 1) |
|---|---:|---|---|
| M-Local | 31 | Municipality-level data with original features | Local |
| M-Global | 12 | Municipality-level with only global features | Global |
| G-Global | 12 | Grid-level with only global features | Global |
| G-Global+ | 18 | Grid-level with global and additional features | Global+ |
| 2SG-Global+ | 18 | Grid-level with global and additional features using a 2-stage classifier | Global+ |
| | | The model is explained in more detail in Figure 8 | |
| M-Naive | 0 | Municipality-level naive baseline based on the target variable average of the training set | None |
| G-Naive | 0 | Grid-level naive baseline only using the target variable average of the training set | None |

– M-Local: Municipality-level data with a subset of the original features ("Local" in Table 1).

– M-Global: Municipality-level with only "global features" ("Global" in Table 1).

– G-Global: Grid-level with only "global features" ("Global" in Table 1).





**Figure 7.** Correlation matrix before removing highly correlated features in the municipality dataset.

– G-Global+: Grid-level with "global" and additional features ("Global+" in Table 1).

   – 2SG-Global+: Grid-level with "Global" and additional features ("Global+" in Table 1) using a 2-stage classifier, explained in more detail in Figure 8.

   – M-Naive: Municipality-level naive baseline that only uses the average of the target variable in the training set at the municipality level.

– G-Naive: Grid-level naive baseline that only uses the average of the target variable in the training set at the grid level.



We begin our experiments by re-implementing the original 510 model that uses municipality-level features, but with feature selection, we call this model M-Local. Not all these features are globally available, so we test this model on a "global" subset of features (M-Global). We then use the features at the grid level and test them in this new resolution (G-Global). In an attempt to improve this model, we take two steps. First, we introduce additional global features called "Global+" and test them in

G-Global+. Second, we implement a 2-stage classifier that handles high-damage data points separately (2SG-Global+). This is another attempt to deal with the high skewness of our target variable.

The flow diagram of this final hybrid model (2SG-Global+) is illustrated in Figure 8. For this model, we first build a binary XGBoost classifier to separate high and low-damage areas (using a 10% damage threshold) with undersampling (using 0.1 as the parameter, reducing the majority class to a size ten times larger than the minority class)[18] to enhance the classification

performance by minimizing the false negatives. Then, we train a second XGBoost regression model (XGBoost-highDamage) using only training data from the high-damage areas. The final result of the 2SG-Global+ model is then (based on the outcome of the binary classifier) either given by the G-Global+ model for data classified as potentially low-damage or by the XGBoost-highDamage model for the rest of the data classified as potentially high-damage.

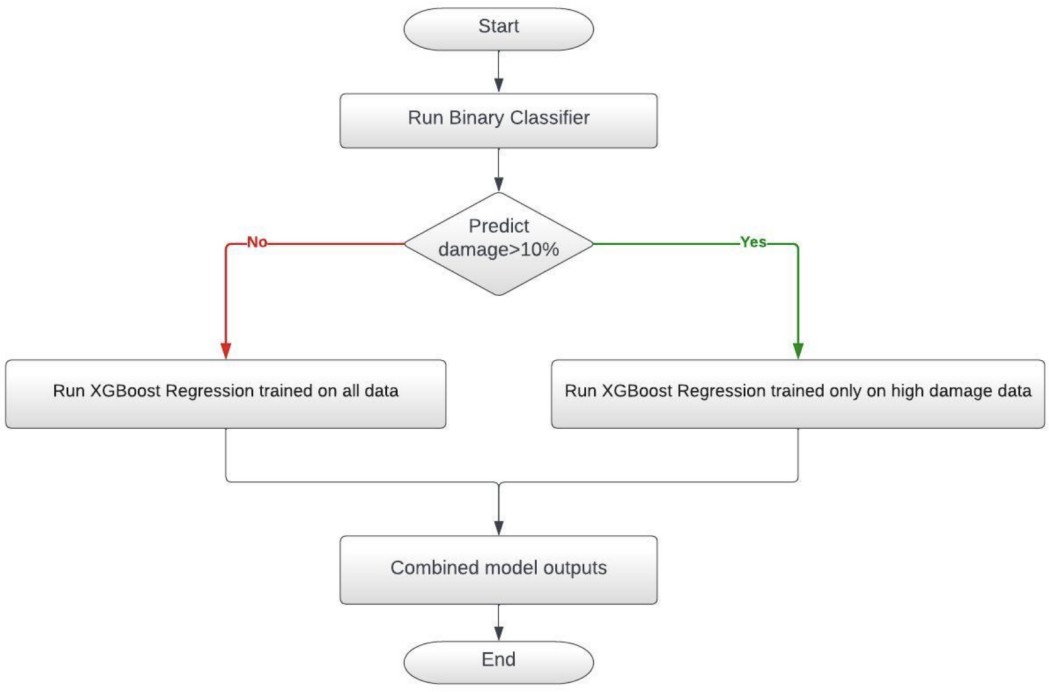

**Figure 8.** Flow chart of combined-model.

---

[18]We also experimented with other parameter settings as well as oversampling strategies (data not shown). However, the results were slightly worse.





## 3.6 Model Evaluation

We perform several types of evaluations using the following error metrics: Root Mean Square Error (RMSE) and Average Error (predicted damage - real damage). We report mean and standard deviation over 20 experiments in each case to average over the variability of the underlying algorithms and sample selections. To have a fair comparison between the models, we evaluate at the municipality level (transforming the results back from the grid level if needed) and only for the data points present in the original municipality data set used by the 510 model.

We first evaluate all the models mentioned in Section 3.5 with a train-test split ratio of 80:20 and stratification as explained in Section 3.1. Note that because the split does not consider the typhoons, (different) data points from the same typhoon may be included in both test and train sets. The rationale behind starting with random train/test splits to evaluate our models is that the more realistic case of typhoon-based train/test splits leads to considerable variability in the performance of the models between different runs, as the severity of typhoons is very heterogeneous. This makes it difficult to assess and compare the performance

of different models. Furthermore, random train/test allows the stratification of test and training sets by severity bins, achieving more stable results. We can thus better analyze and compare the efficiency of different model types, the feature importance, and the impact of changing from a municipality to a grid-based model.

However, to understand our best-performing models' performance in a real-time use case, we also undertake a walk-forward evaluation and leave-one-out cross-validation (LOOCV) wherein the typhoon timings are preserved. The walk forward evalu-

ation uses a chronologically ordered set of typhoons, starting with an initial training set of 27 typhoons (approx. 70% of the data). Each iteration adds a new typhoon to the training set, and the model is tested on the next one (making for 12 iterations for each of the 12 remaining typhoons). The aim is to determine how well the model learns from older typhoons' characteristics to predict the next. We implemented an alternative version where the oldest typhoon is dropped when adding a new one (making the training window fixed), but as the results were statistically the same, we do not report them here. In the LOOCV, we cycle

through all typhoons, using one typhoon as a test set and the others to train the models. This setting makes more data available (allowing the use of "future" data) for training than the walk-forward scenario.

## 3.7 Model Explainability

We employed SHAP (SHapley Additive exPlanations), a game theory approach developed to explain the contribution of each feature to the final output of any ML model (Lundberg and Lee, 2017b). SHAP values provide global and local interpretabil-

ity, meaning we can assess how much each predictor and observation contributes to the classifier's performance. The local explanations are based on assigning a numerical measure of credit to each input feature. Then, global model insights can be obtained by combining many local explanations from the samples (Lundberg et al., 2019). As mentioned by the authors, the classic Shapley values can be considered "optimal" in the sense that within a large class of approaches, they are the only way to measure feature importance while maintaining several natural properties from cooperative game theory (Lundberg and Lee,

2017a). SHAP's output helps to understand the general behavior of our model by assessing the impact of each input feature in the final decision, thus enhancing the usefulness of our framework.





## 4 Results

This section presents the results from the evaluations described in the previous section. We start with random train/test splits that do not stratify by typhoons. Then, we show the LOOCV and walk-forward settings and finish with an illustrative case study.

### 4.1 Random Test-Train Split Evaluation / Regression Model Performance

Table 3 shows the RMSE (and standard deviation over 20 runs) for the four regression models and the two-stage hybrid model described in Section 3.5, as well as the two baseline models which predict the average damage in a municipality/grid as seen in the training data (ignoring all features). We present the results per damage bins and the average over all the test data, which we refer to as the weighted average (recall that the data is heavily skewed toward the first few bins). Also note that when we compute the metric, the predictions of the grid-based models are converted to the municipality level, and only municipalities present in the original data used by the 510 model are considered for a fair comparison across all models.

First, we note that the RMSE score increases proportionally to the bin's interval, meaning those municipalities that experienced more damage also have higher errors associated with the model's prediction. As previously stated, we will face a higher standard deviation in the latest bins since the dataset has more data points in the initial bins. When we limit the variables from the M-Local model to only those globally available (down to 12 features) in M-Global, the model's performance does not suffer significantly in terms of RMSE. Further, as we add features to our set (G-Global+), the performance improves slightly in the higher damage range when compared to G-Global. Finally, the two-stage model (2SG-Global+) achieves the best RMSE for bin 4, as it is designed to perform slightly worse for the low-damage bins, which results in a weighted average RMSE of 4.73. The table also includes two baselines: one for the municipality and one for the grid level. These achieve a worse performance compared to the proposed models. They perform the best in the middle bin, as the overall average falls into this bin.

Additionally, Table 4 shows the average error achieved by the same models. This is the average difference between estimated and actual damage values, so the model tends to underestimate the real damage when the average error is negative. As we can see, for the five models under analysis, the real damage is overestimated for the first three bins and is underestimated for the last two with the highest damage. This effect is expected due to the skewness of the data. While the average error over all bins (weighted average) remains close to 0 for the municipality-level models, introducing the grid level increases the models' tendency to underestimate. However, on average, the 2SG-Global+ model corrects the overall bias down closer to 0. In particular, we again notice a significant improvement in bin 4.

Next, we explore the feature importance of the globally available variables (column Global+ in Table 1). Figure 9 shows the beeswarm plot of the SHAP values for the G-Global+ model. It allows us to observe the impact of each feature on the model output. For instance, among the most critical variables, high values (red points) of the wind speed feature indicate a high positive contribution to the prediction (positive SHAP value). High values of the 6-hour maximum rainfall (in red) are positively associated with damage, while lower ones (in purple) have a negative one. This effect gets diluted when the rainfall aggregation is for 24 hours, where we can also observe an increased negative impact of high feature values. Interestingly,



**Table 3.** Table of RMSE per bin and the weighted average for the five proposed models and two baselines (standard deviation over 20 runs in parentheses).

| Bin interval | M-Local | M-Global | G-Global | G-Global+ | 2SG-Global+ | M-Naive | G-Naive |
|---|---|---|---|---|---|---|---|
| 1. [0, 0.00009] | 0.27 (0.05) | 0.35 (0.11) | 0.27 (0.10) | 0.24 (0.08) | 0.39 (±0.20) | 2.22 (0.01) | 0.83 (0.00) |
| 2. (0.00009, 1] | 2.10 (0.34) | 2.30 (0.27) | 1.60 (0.16) | 1.50 (0.13) | 1.94 (0.17) | 2.03 (0.02) | 0.68 (0.01) |
| 3. (1, 10] | 4.37 (0.46) | 4.45 (0.54) | 4.57 (0.48) | 4.63 (0.54) | 5.64 (0.59) | 2.82 (0.16) | 3.69 (0.12) |
| 4. (10, 50] | 13.47 (1.10) | 14.58 (1.01) | 14.27 (1.05) | 14.02 (0.66) | 12.48 (0.90) | 24.81 (1.15) | 25.34 (0.92) |
| 5. (50,100] | 27.93 (4.98) | 30.54 (5.69) | 33.51 (3.89) | 31.62 (3.50) | 31.67 (3.97) | 60.88 (3.39) | 63.97 (2.35) |
| Weighted Average | 4.42 (0.29) | 4.77 (0.32) | 4.82 (0.28) | 4.71 (0.20) | 4.73 (0.28) | 8.03 (0.24) | 8.25 (0.24) |
| Total Features | 31 | 12 | 12 | 18 | 18 | 0 | 0 |

**Table 4.** Table of Average Error per bin and the weighted average for the five proposed models and two baselines (standard deviation over 20 runs in parentheses).

| Bin interval | M-Local | M-Global | G-Global | G-Global+ | 2SG-Global+ | M-Naive | G-Naive |
|---|---|---|---|---|---|---|---|
| 1. [0, 0.00009] | 0.08 (0.01) | 0.08 (0.02) | 0.03 (0.01) | 0.03 (0.01) | 0.04 (0.01) | 2.22 (0.01) | 0.83 (0.00) |
| 2. (0.00009, 1] | 0.89 (0.07) | 1.12 (0.09) | 0.63 (0.06) | 0.58 (0.03) | 0.67 (0.05) | 2.01 (0.02) | 0.64 (0.01) |
| 3. (1, 10] | 0.89 (0.28) | 0.94 (0.34) | 0.00 (0.28) | 0.10 (0.25) | 1.00 (0.35) | −1.42 (0.16) | −2.80 (0.12) |
| 4. (10, 50] | −6.28 (1.32) | −7.16 (1.36) | −7.13 (1.00) | −6.66 (0.92) | −4.53 (0.97) | −21.77 (1.18) | −22.57 (0.81) |
| 5. (50,100] | −20.55 (6.12) | −24.57 (6.40) | −25.20 (3.93) | −23.49 (3.52) | −25.39 (3.97) | −59.97 (3.03) | −62.72 (2.16) |
| Weighted Av. | 0.01 (0.10) | 0.02 (0.11) | −0.33 (0.06) | −0.30 (0.06) | −0.06 (0.09) | −0.03 (0.07) | −1.43 (0.05) |
| Total Features | 31 | 12 | 12 | 18 | 18 | 0 | 0 |

the track distance can have a positive and negative impact, especially when its values are low. Furthermore, historical data (percent_houses_damaged_5years) mostly positively impacts the prediction. The elevation feature (TOP_mean_elevation_m) does not provide a clear picture alone, while the mean slope feature shows that flat areas are more likely to receive damage than others. We also observe a positive impact of the coast length feature on damage estimation since coastal areas are more prone to storm surges and landslides.

Further, social-demographic features provide a window into the unequal distribution of damage caused to the population. Variables concerning total_houses, urban, and rural measures indicate that the areas with fewer houses (less urbanization) are affected more than those in the cities. The relative wealth index (RWI) also shows that those affected tend to come from economically disadvantaged areas. In summary, the most critical features involve the characteristics of the typhoon in terms of wind and precipitation. Although contributing less to the model performance, the conditions on the ground, such as population, urbanization type, or the relative wealth index (RWI), provide a clear directional signal on how they influence the expected damage.





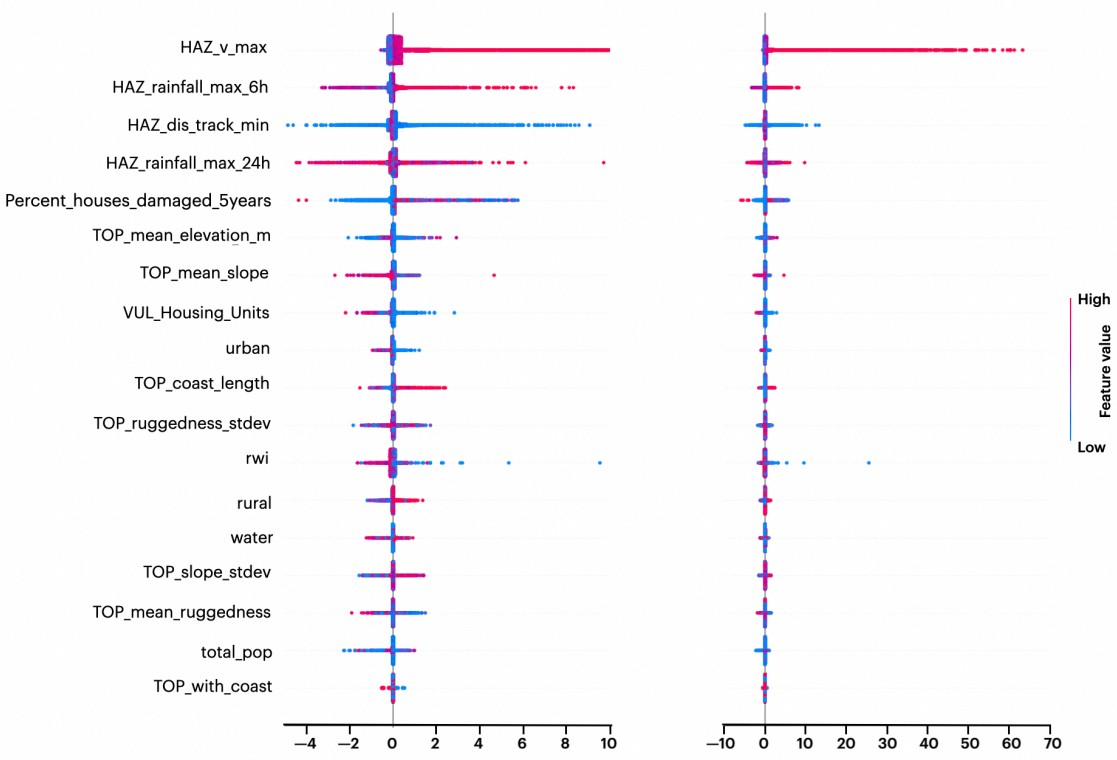

**Figure 9.** SHAP values for variables in G-Global+ model, sorted by importance of all the globally available features. The right panel shows the whole scale of SHAP values while the left panel shows a reduced x-axis range for better data visualization.

## 4.2 Evaluation by Typhoon

In this section, we now evaluate our best model (2SG-Global+) in a more realistic setting with train/test splits with stratification by typhoon in two ways: iterative walk-forward evaluation and leave-one-out cross-validation (LOOCV). Figure 10 gives a

350 graphical representation of these two typhoon-based stratification strategies explained in detail in Section 3.6. For the sake of comparison, we show the LOOCV evaluation here only for the same 12 typhoons used for the walk-forward evaluation – this makes these results different from those in the previous section. The overall performance is slightly better in LOOCV compared to the walk-forward setting, which makes sense, as it has access to more training data.

Table 5 shows the RMSE and Table 6 the average error for the 2SG-Global+ model, the M-Local model and G-Naive

355 model (i.e. the average historical damage), evaluated using the two methods. Compared to the G-Naive, our hybrid 2SG-Global+ model achieves substantially better RMSE in the higher damage bins, especially at the highest range of (50, 100], having RMSE of 33.61 (±22.30), compared to 57.02 (±6.19) of the G-Naive (LOOCV evaluation). Additionally, the weighted average RMSE for the walk-forward and LOOCV scenarios of the 2SG-Global+ model are 2.55 (±1.98) and 2.48 (±1.93), which indicate a better performance than the M-Local model with RSME of 2.74 (±1.66) and 2.64 (±1.66) respectively. For





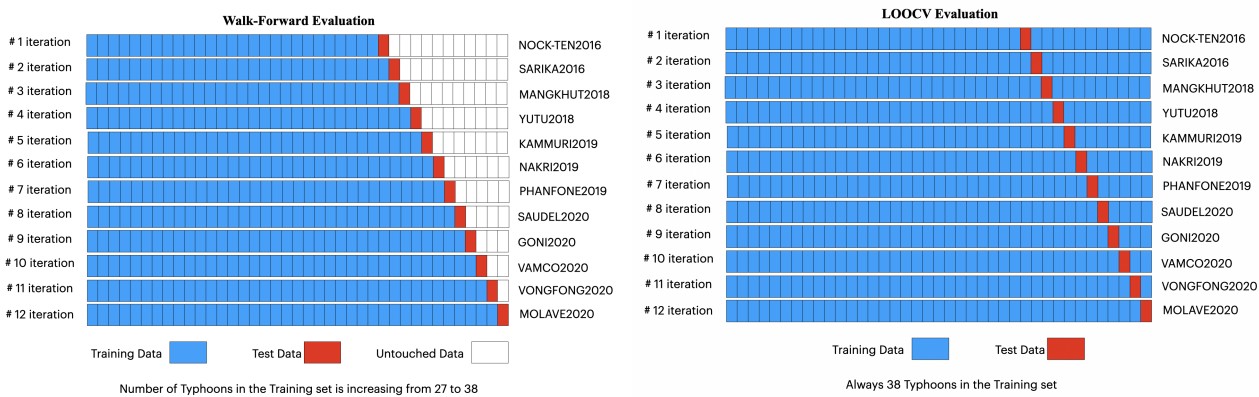

**Figure 10.** Schematic diagram of the Walk Forward (left) and LOOCV Typhoon-based evaluation (right) to illustrate how the dataset is split into the training and test sets.

the first three bins, the 2SG-Global+ model has the lower RMSE, while for the last two bins (high-damage data), the M-Local performs slightly better than the 2SG-Global+ model. Furthermore, the proposed model, apart from the highest damage bin, improves the average error.

Figures 11 and 12 show the performance of the two models only in grid cells with damage > 10, evaluated using LOOCV (calculated for all typhoons having such high damage impact, instead of just the 12 used in Tables 5 & 6). We find that, for most typhoons, the error is substantially reduced. The average error tells us that the 2SG-Global+ model's predictions are less pessimistic; that is, the model underestimates the damage to a lesser extent than what we would have predicted based on the historical baseline (G-Naive) and, in some cases, even overestimates it. In general, the G-Naive baseline model has the

**Table 5.** RMSE per bin and the weighted average for M-Local, 2SG-Global+ and G-Naive models (with the same size of dataset) in walk-forward and leave-one-out c.v. (LOOCV) evaluation.

| Bin interval | M-Local | | 2SG-Global+ | | G-Naive | |
|---|---|---|---|---|---|---|
| | Walk-forward | LOOCV | Walk-forward | LOOCV | Walk-forward | LOOCV |
| 1. [0, 0.00009] | 0.22 (0.16) | 0.25 (0.17) | 0.10 (0.07) | 0.11 (0.08) | 0.91 (0.03) | 0.84 (0.01) |
| 2. (0.00009, 1] | 1.89 (1.50) | 1.82 (1.41) | 1.32 (1.84) | 1.37 (2.08) | 0.76 (0.08) | 0.69 (0.08) |
| 3. (1, 10] | 5.61 (3.12) | 5.16 (2.32) | 4.82 (3.42) | 4.86 (3.45) | 3.41 (1.10) | 3.47 (1.12) |
| 4. (10, 50] | 11.24 (2.67) | 10.41 (4.58) | 12.89 (3.08) | 12.32 (3.08) | 16.33 (3.94) | 16.40 (3.93) |
| 5. (50, 100] | 29.39 (16.72) | 33.24 (11.17) | 31.05 (21.92) | 33.61 (22.30) | 56.95 (6.24) | 57.02 (6.19) |
| Weighted Average | 2.74 (1.66) | 2.64 (1.66) | 2.55 (1.98) | 2.48 (1.93) | 2.85 (2.00) | 2.84 (2.04) |
| Total Features | 31 | | 18 | | 0 | |





**Table 6.** Average Error per bin and the weighted average for M-Local, 2SG-Global+ and G-Naive models (with the same size of dataset) in walk-forward and leave-one-out c.v. (LOOCV) evaluation.

| Bin interval | M-Local | | 2SG-Global+ | | G-Naive | |
|---|---|---|---|---|---|---|
| | Walk-forward | LOOCV | Walk-forward | LOOCV | Walk-forward | LOOCV |
| 1. [0, 0.00009] | 0.07 (±0.08) | 0.10 (0.09) | 0.02 (0.05) | 0.02 (0.05) | 0.91 (0.03) | 0.84 (0.01) |
| 2. (0.00009, 1] | 1.00 (1.02) | 1.10 (1.17) | 0.51 (1.03) | 0.67 (1.40) | 0.71 (0.11) | 0.63 (0.11) |
| 3. (1, 10] | 2.07 (2.71) | 1.84 (2.26) | 0.69 (3.17) | 0.95 (3.34) | −2.68 (0.97) | −2.76 (0.99) |
| 4. (10, 50] | −4.82 (6.83) | −4.16 (6.95) | −2.89 (9.83) | −3.06 (9.40) | −15.27 (3.18) | −15.36 (3.17) |
| 5. (50, 100] | −29.39 (16.72) | −33.24 (11.17) | −31.05 (21.92) | −33.61 (22.30) | −56.95 (6.24) | −57.02 (6.19) |
| Weighted Average | 0.35 (0.58) | 0.35 (0.57) | 0.03 (0.87) | 0.07 (0.84) | −0.15 (0.98) | −0.22(0.99) |
| Total Features | 31 | | 18 | | 0 | |

worst RMSE. It also has the worst Average Error for all typhoons compared to the two other models. For some typhoons like Meranti16 and Vamco20, the M-Local model has the lowest RMSE, while for the earlier typhoons such as Haiyan13, Utor13, Rammasun14, and a few more, the performance of 2SG-Global+ is very similar to M-local. It was even better in some cases, like Bopha12 (the most severe), which is remarkable given that M-Local includes 31 and 2SG-Global+ only 18 features.

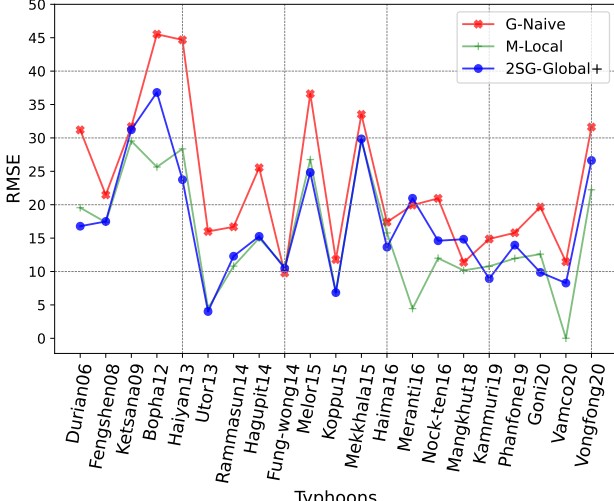

**Figure 11.** RMSE for areas with damage>10% by typhoon (LOOCV, only typhoons with high-damage areas shown).

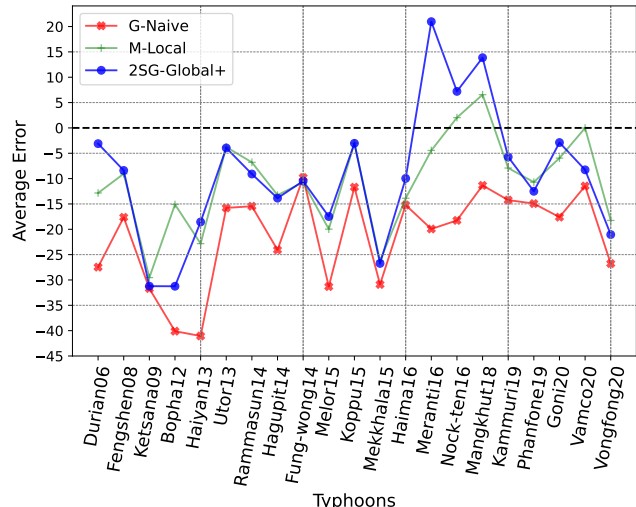

**Figure 12.** Average error for areas with damage>10% by typhoon (LOOCV, only typhoons with high-damage areas shown).





**Table 7.** Performance in predicting municipalities with damage>10% tested using LOOCV. True Positive (TP), False Positive (FP), False Negative (FN), Precision, Recall, and F1 scores for 2SG-Global+ (average of 20 runs), M-Local Model, and the G-Naive baseline.

|  | TP | FP | FN | Precision | Recall | F1 |
|---|---|---|---|---|---|---|
| G-Naive baseline | 0 | 0 | 447 | 0 | 0 | 0 |
| M-Local | 254 | 173 | 193 | 0.595 | 0.568 | 0.581 |
| 2SG-Global+ | 295 | 169 | 152 | 0.636 | 0.660 | 0.648 |
| improvement | 16.14% | −2.31% | −21.24% | 6.89% | 16.17% | 11.48% |

### 4.3 Action Trigger Application

The above comparison, however, needs to reflect the usefulness of these models in decision-making during emergencies. The real-world application case would be to predict municipalities where the damage will be more significant than 10% so an appropriate action can be triggered. Using the output of our models for this classification task, we show in Table 7 the corresponding number of True Positive (TP), False Positive (FP), False Negative (FN), as well as Precision (P), Recall (R), and F1 scores for the G-Naive baseline and the average of 20 runs of the 2SG-Global+ and the M-Local models. The G-Naive model never predicts the damage will be over 10%, as most of the data skews to minor damage. This results in a low RMSE because, indeed, for most data points, this prediction is correct but would never trigger appropriate action. Alternatively, the M-Local identifies the municipalities having damage over 10% with a precision of 0.60 and a recall of 0.57. The 2SG-Global+ model then improves this performance by increasing the number of true positives and decreasing the number of false positives and false negatives. This results in a 6.9% Overall, the F1 measure of the proposed model is the best at 0.65 (an 11.5% improvement).

In a thought experiment, if decision-makers would use the output of 2SG-Global+, instead that of M-Local, 41 more municipalities (an increase of 16.1%) experiencing greater than 10% severely damaged houses would receive relief (more true positives, fewer false negatives), and at the same time Four fewer municipalities (a decrease of 2.3%) with damage not exceeding the threshold would not receive aid, saving resources (fewer false positives).

In summary, in this real-world scenario, our model improves resource allocation by targeting better the affected areas where early actions or AAs will be deployed, increasing the correctly predicted damaged areas and reducing false alarms.

### 4.4 Case study

To illustrate the behavior of the 2SG-Global+ model and compare it with the M-Local model, we visualize the prediction results at the municipality level for a single typhoon estimated by these two models. We choose Melor, a powerful typhoon of category 4 that struck the Philippines in December 2015. Figure 13 shows the typhoon track (orange line) and actual damage at the municipality level during this typhoon. To have a fair comparison between the 2SG-Global+ and the M-Local





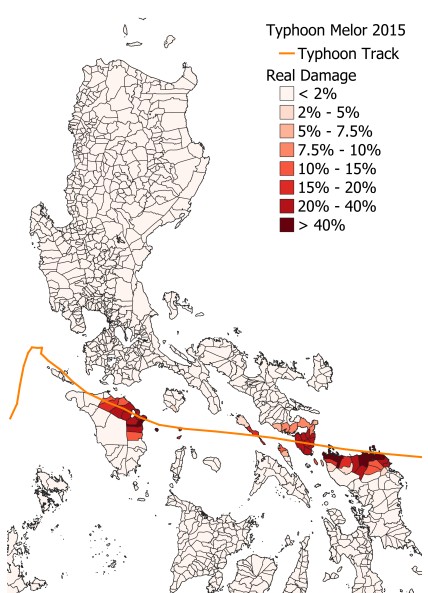

**Figure 13.** The real damage per municipality as reported for Typhoon Melor in 2015.

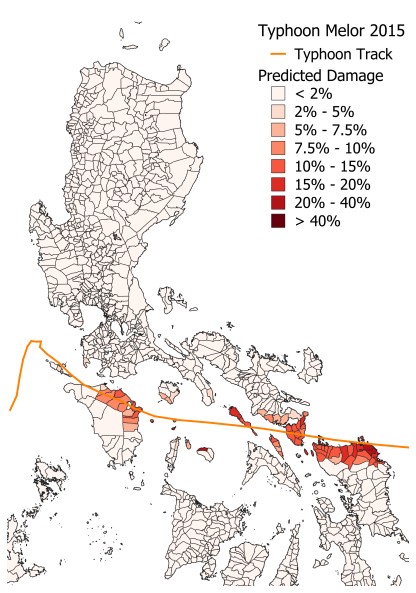

**Figure 14.** The damage as predicted by the 2SG-Global+ model for Typhoon Melor aggregated from the 0.1-degree grid to the municipality level.

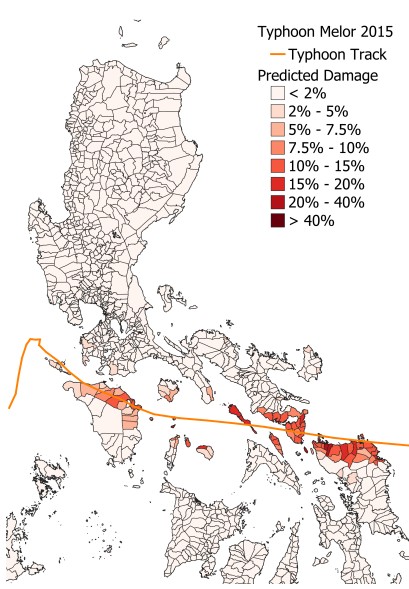

**Figure 15.** The damage as predicted by the M-Local model for Typhoon Melor.

395 models, we transformed the results of the 2SG-Global+ from the grid level to the municipality. We used only the data from the municipalities present in the original municipality data set used by the 510 model.

Figures 14 (for 2SG-Global+) and 15 (for M-Local) show the predicted damage, and Figures 16 and 17 the corresponding errors of the two models for the Melor typhoon. For individual municipalities, the models either underestimate the damage (blue squares) and, in a few cases, also overestimate it (red), but in general, the areas with predicted damage coincide or are 400 close to the affected zones. The average error of 2SG-Global+ for Melor is approximately -0.5, slightly underestimating the impact of the typhoon on average, while for the M-Local model, this error is considerably larger (-2.98).

Still, we generally observe very similar values, with little noticeable differences between the two models in these figures. However, going back to Figures 11 and 12 we do observe that the 2SG-Global+ model performs slightly better with a reduction of the RMSE by 1.9 (-7.2%) and the average error by 2.5 ( -12.5%) in the high damage areas (municipalities with damage 405 >10%). This was also observed in the action trigger application analyzed in the previous subsection, where we found that for the Melor typhoon, the 2SG-Global+ model correctly identified 35 municipalities as highly damaged (only missing 8). In contrast, the M-Local model only identified 25 of them.




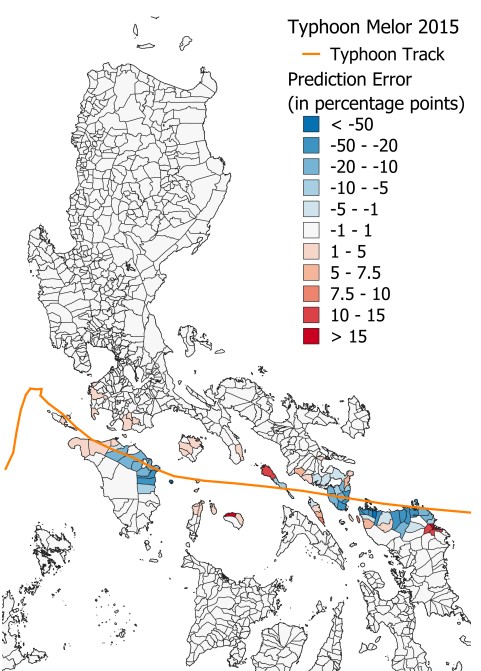

**Figure 16.** The error of the 2SG-Global+ Model aggregated by municipality (predicted - real).

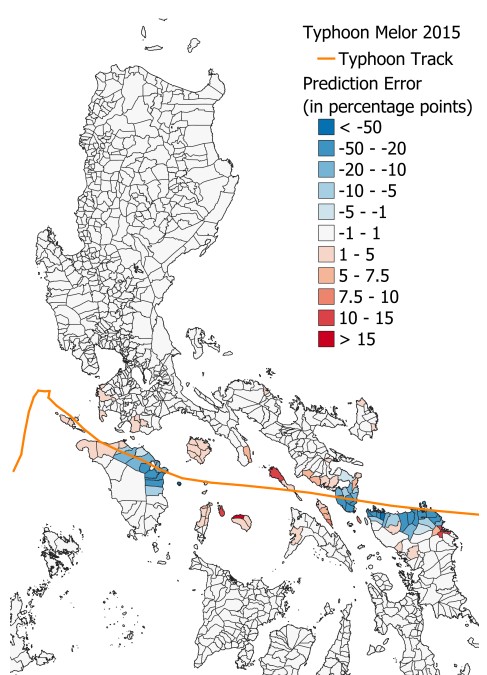

**Figure 17.** The error of the M-Local Model per Municipality (predicted - real).

## 5 Discussion & Conclusions

In this study, we created 2SG-Global+, a grid-based typhoon damage prediction model for the Philippines. It uses fewer features than the original municipality-based 510 model (Teklesadik et al., 2023) while conserving performance in terms of RMSE. The features of our grid-based model have all been selected from an open-access global database, which facilitates the possibility of extending the grid-based model to different geographical contexts. When applied as a classification model for high damage areas (more than 10% of housings destroyed) and despite excluding hazard and vulnerability features only available in the Philippines, our hybrid 2SG-Global+ model achieves an F1 = 0.648, which surpasses a variant of the 510 model (M-Local) with F1 = 0.581. Our model shows a higher True Positive rate (295 instead of 254 of M-Local) and a lower False Negative rate (152 instead of 193). The 2SG-Global+ model even slightly outperforms M-Local, respecting the False Positive rate (169 instead of 173), meaning fewer resources would be wasted. Furthermore, the availability of the model results in a $0.1°$ grid, allowing for a more targeted AA approach.

In the Philippines, at the moment this manuscript is written, there is a partly human-out-of-the-loop (510 model to select municipalities) and a partly human-in-the-loop approach (barangay validation committee that selects the program participants) (van den Homberg et al., 2020) in place for AA. With a grid-level model, the barangays (smaller than municipalities) can be better targeted by aggregating the grid cells into barangays instead of the municipality level. It should be noted that the grid-cell model can distribute some of its predictions to municipalities outside the municipalities of the target training data, with two



consequences. It allows the uncovering of areas that might have been overlooked in the data collection (e.g., areas with mild damage that have not been reported but are helpful to train the model). However, it also might allocate damage to areas of minor importance (e.g., neighboring villages with very few houses).

Our study demonstrates that creating a grid-level impact-based forecasting model using global features for the Philippines is possible. Further research will focus on applying the grid-level approach to other TC-prone countries so that the replicability of the model can be tested. Whereas data on the features of the grid-based model are available for other countries from global repositories, this will not be the case for the target data. The aggregation level of the target data available for other countries will differ. This data might be more detailed in some data-richer countries, making the model less sensitive to disaggregation by using building footprint data. Also, our geospatial workflow could be audited for biases (Masinde et al., 2023). Efforts are underway to identify biases within satellite building datasets like OSM, Google, and Bing Building Footprint to specific attributes such as vulnerability (Gevaert, 2022). It will be essential to assess the sensitivity of the grid-level model to these biases. In our research, the target data has been damage to houses, but damage to other assets can be used. For example, Boeke et al. (2019) and van Brussel (2021) developed a model for predicting damage to rice fields in the Philippines initially at the province level and later at the municipality level. Damage to rice fields will require a different way of disaggregating. Land-use/land-cover data could be used to assign rice damage to a specific grid cell.

The models in our research are trained with observed TC data. It is important to emphasize that the performance of an operational impact-based forecasting model is determined not only by the performance of the model itself but also by the forecast skill of the real-time hazard forecast that goes into the model. For example, the position error for three days of European Centre for Medium-Range Weather Forecasts (ECMWF) ensemble forecasts averaged 150–200 km over the last few years (MacLeod et al., 2021). Also, the Philippine Red Cross requires 72 hours to implement early actions such as distributing house-strengthening kits. However, in the Philippines, of the 522 TCs that made landfall from 1951-2020, 146 TCs (28%) underwent rapid intensification, defined as the upper 95th percentile increase of TC maximum winds in a 24-hour period. Of this 28%, 82% had at least typhoon intensity (Tierra and Bagtasa, 2023; Fudeyasu et al., 2018). This means that the threshold level of any impact-based forecasting model will not be reached at 72 hours. The operational 510 model uses the forecasts from ECMWF as they represent the state-of-the-art in TC forecasting (MacLeod et al., 2021) and can feed directly into an automated workflow. However, the forecasts from the Philippine Atmospheric, Geophysical, and Astronomical Services Administration (PAGASA, 2023) are contextualized with their local knowledge, and updates are often available faster. Therefore, further research will explore the adoption of the PAGASA forecasts. When replicating the grid-level model to other countries, a forecast skill assessment of the different forecasts available for that country has to be done.

Apart from the primary hazard, a possible future improvement of our model may come in the form of dynamic modeling of the consecutive or secondary hazards caused by a TC. For example, for storm surge, dynamic models, such as the Global Tide and Surge Model, are being developed (Bloemendaal et al., 2019). Incorporating these dynamic models into an ML model might improve the secondary hazard features used in grid- and municipality-based models. The new technologies being developed as part of Digital Earth (Annoni et al., 2023) can play a role here.



With our novel model, we are able to increase the number of true positives so that decision-makers can distribute the limited resources better to those who will suffer from more damage. Apart from considering the purely technical performance of our grid-level model, there are opportunities and challenges in adopting an artificial intelligence-based model by decision-makers (Kbah and Gralla, 2023). A clear opportunity is that the grid-level model uses features that can be the same across countries, which will benefit consistency and comparability. Also, a grid-level model based on global features can be more easily rolled out to new countries, requiring less time and resources. A challenge might be that models based on artificial intelligence are more of a black box to users with limited data and digital training than expert or rule-based trigger models. For example, setting a threshold might be less intuitive for an xgboost model than for a model primarily based on wind speed. The Red Cross National Societies in Bangladesh and Mozambique currently use, for example, a relatively straightforward trigger model based on combining a wind speed forecast with vulnerability information (Sedhain et al.). Bierens et al. (2020) explain the importance of an impact-based forecasting model's legitimacy, accountability, and ownership. The knowledge exchange between the developer and the end users of the models falls short if it is just seen as a matter of technology transfer. Instead, co-creation is essential to incorporate the end user's needs fully.

To conclude, relying on globally available data sources and working at the grid level holds the potential to render an ML-based impact model generalizable and transferable to locations outside of the Philippines. The grid-level model can contribute to developing an impact-based forecasting model in a country that still needs to develop a local one. The long-term adoption of our model based on AI may take place by forming an additional source of information next to more expert-based or local models of a government's AA pipeline. Future research will focus on the validation of the model in other countries. Also, UN OCHA and the Red Cross Red Crescent Movement aim to gain experience by running the grid-level model parallel to existing trigger models. Expanding the application of this transferable TC model to other countries will facilitate the scaling up of anticipatory action for TCs.

**Code/Data Availability**

We make the original and processed data available at http://rb.gy/f27wy, and the code is available at https://github.com/rodekruis/GlobalTropicalCycloneModel.



*Author contributions.* Conceptualization: M. van den Homberg, L. Milano, K.Kalimeri, D. Paolotti, ML. Turner; Data curation: A. Tekle-sadik, P. Ndirangu; Investigation: M. Kooshki, A. Kaltenbrunner, Y. Mejova; Writing – original draft preparation: M. Kooshki, A. Kaltenbrun-ner, Y. Mejova; Writing – review & editing: M. Kooshki, M. van den Homberg, K. Kalimeri, A. Kaltenbrunner, Y. Mejova, P. Ndirangu, D.
Paolotti, ML. Turner. All authors reviewed the results and approved the final version of the manuscript.

*Competing interests.* The authors declare no competing interests.

*Acknowledgements.* The authors acknowledge support of Mersedeh Kooshki Forooshani from the Lagrange Project of the Institute for Scientific Interchange Foundation (ISI Foundation) funded by Fondazione Cassa di Risparmio di Torino (Fondazione CRT). KK, YM and DP acknowledge financial support from the Lagrange Project of the Institute for Scientific Interchange Foundation (ISI Foundation) funded
by Fondazione Cassa di Risparmio di Torino (Fondazione CRT). Marc van den Homberg and Aklilu Teklesadik were supported by the Princess Margriet Fund and the Forecast-based financing project with the Philippine Red Cross, funded by the German Red Cross. The model development of the 510 model greatly benefited from the contributions of all the volunteers who supported 510, an initiative of the Netherlands Red Cross, since its start in 2016.



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
