# Peer review of "Towards a Global Impact-based Forecasting Model for Tropical Cyclones"

_EGUsphere, 2023_

## Author Response (AR1)

**Response to Reviewer Comments**

"Towards a global impact-based forecasting model for tropical cyclones"

November 24, 2023

We thank the reviewers for their helpful comments. The replies to the comments can be found in the interactive community forum, but we also provide the responses below. We also submit an updated version of the manuscript with the changes highlighted.

**RC1, Guido Ascenso, 11 Oct 2023**

Line 146: what is the definition of "fully damaged"?
A: The definition of damage comes from the Department of Social Welfare and Development that we cite in Section 3.1. The exact definition of the extent of damage is subject to interpretation of the assessors and the municipality. We assume that this definition does not vary between municipalities, but it is true that it may present a noise in the data.

Line 158: towards the end of the line the number of the figure is missing (should be Figure 1, presumably).
A: Apologies for not catching this broken reference. It should be referring to Figure 1. We will fix it in the next version of the paper.

Lines 235-240: the authors state that only variables with correlation > 0.99 were removed. However, figure 7 shows several more features with also quite high correlation, and some with correlation that seems close to -1, which is just as significant. Rather than using a fixed (too high, in my opinion) threshold on correlation, I would prefer a feature selection algorithm, even a basic one.
A: We chose the cutoff of feature correlation at absolute values larger than 0.99 to make sure that whole feature groups were not removed. We will clarify in the next version that we used the absolute value in the next version of the paper. The largest negative correlation between the remaining variables is -0.85 between HAZ_dis_track_min and HAZ_v_max. Further, we also experimented with the VIF method for removing additional features, but this decreased the performance of the model. Ultimately, the model we chose – XGBoost – is robust to multicollinearity, thus we chose to include more variables in order to let the model choose the most important ones, while still being able to have some performance gains through including the highly correlated variables.

Lines 325-327: "While the average error over all bins (weighted average) remains close to 0 for the municipality-level models, introducing the grid level increases the models' tendency to underestimate. However, on average, the 2SG-Global+ model corrects the overall bias down closer to 0". First of all, what do the authors mean by "remains close to 0"? Doesn't table 3 report that even the best model (M-Local) has a RMSE of 4.42 on the weighted average data? Is the phrase 'close to 0' supposed to me 'is the lowest of the models considered'? Otherwise, it's too subjective. Similarly, the sentence „the 2SG-Global+ model corrects the overall bias down closer to 0", this is not reflected by the results in Table 3, as the G-Global+ model has lower weighted average RMSE error than the SG-Global+ model.
A: In lines 325-327 we are speaking about the results shown in Table 4, on the average error, which measures the difference between estimated and actual damage values. This measure, overall, is very close to 0 for the first two municipality-based models (first two columns of the table), and becomes more negative for the grid-based models (third through fifth columns). Both, "Average Error" and "Root Square Mean Error" measure error, but whereas the first preserves the direction of the error, the second does not. Thus, our AE results are distinct from those measured using RMSE in Table 3.

Table 3: in the text, it's commented how the 2SG-Global+ model is the best performing one; this is then repeated in line 348. However, Table 3 shows that the M-local model was the best performing one for the 5th bin, which to me seems like the most important one, as it is the one with the most significant damages. Is the argument that bin 5 has too few data points, and therefore the results obtained on it are less statistically robust than those for the other bins? If that's the case, I would be interested in seeing follow-up studies applied to areas that have more data for this bin, to see if the conclusions drawn here stand true.

A: It is true that the M-local model has the best performance on Bin 5 – that with the most damage. However, 2SG-Global+ is as we correctly state in line 319 the best performing model in bin 4. This bin is equally if not more important than bin 5. It is equally important as it also covers areas with damages larger than 10% which would contribute to the triggering of Anticipated Action. It is more important as the number of damaged areas in this bin is significantly larger, overall if bins 4 and 5 were merged the weighted error would be 15.18 for 2SG-Global+ vs 15.53 for M.local, showing the better performance of 2SG-Global+ in the high damage bins. In any case, in this study we are not necessarily looking for the best model per se, but it is seeking to achieve a good performance in models that use different sets of features – those that are more available globally, and which make the prediction at a more granular (grid) level. Further, we explore the performance of the models on the high-damage areas in section 4.3 Action Trigger Application, wherein we demonstrate the performance of the models in detecting the municipalities that were in most need of assistance. This experiment showed that the proposed grid-based model resulted in a better true positive, false positive, and false negative rates, which would have improved the use of limited anticipatory action resources. Finally, when we state in line 348 that "we now evaluate our best model (2SG-Global+)" we refer to the best global model we have introduced, not necessarily the best overall model. We will clarify that in subsequent versions of the paper.

**RC3, Anonymous Referee #2, 25 Oct 2023**

This work aligns well with the scope of Natural Hazards and Earth System Sciences. This work improved the spatial resolution and accuracy of the previous typhoon impact forecast model. The technical details of the methods are presented thoroughly, and the results are discussed in a balanced and appropriate manner. Some of my concerns have been addressed by the author's responses to the comments from other reviewers. Below are my additional comments.

1. Line 141: How many 0.1-degree cells do you apply in this study area and models?
A: We use 3726 0.1-degree cells that overlap with land. We will add this information to the next version of the paper.

2. Line 152-154: "Hence, missing damage percentage values in the original data of the municipality-based 510 model were replaced with zero for records with low wind speed (below 25m/s) and rainfall (below 50mm)." Please provide a rationale for choosing these specific thresholds for low wind speed and rainfall.
A: We defined the "low" threshold for both rainfall and windspeed by utilizing long-term average observation. This decision is rooted in the understanding that housing damage is unlikely to occur under conditions of below-average rainfall and wind speed. We will add this clarification in the next version of the paper.

3.Line 158: "To do so, we use the number of buildings from Google Building Footprint data4 (See Figure ?? for a visualization of this data) to compute transformation weights." Can you

clarify what transformation weights are? What methods did you use to convert the damage data at the municipality level to the 0.1 grid?

A: Transformation weights are the proportion of the number of buildings in each grid and municipality intersection with respect to the total number of buildings in each municipality. The number of buildings within each grid in a given municipality and the total number of buildings in a municipality are determined by counting the number of building centroids from Google Open Buildings data and using shape-files for the borders of the municipalities obtained from the Humanitarian Data Exchange (HDX). We are trying to ascertain what proportion of the houses in each municipality should be assigned to the grid. For example, for a municipality wholly contained within a grid, all the houses in the municipality would be assigned to the grid and the weight would be 1. To convert the percentage of damaged houses from the municipality level to the grid level, the number of houses damaged and the total number of houses per municipality are multiplied by the weights to get the total number of houses and houses damaged per grid. The percentage of houses damaged are computed from the two values.

4. Line 159: Does OSM stand for OpenStreetMap? Please spell out the name when it is mentioned for the first time.

A: Yes, OSM stands for OpenStreetMap. We will update this in the next version of the paper.

5. Line 163-164 : "We normalize by the number of buildings in a grid cell to get the back-transformation weights from grid cells to municipalities." Please clarify what "back-transformation weights" are.

A: The back-transformation weights follow the same principle as those of the transformation weights but in the opposite direction, ascertaining what proportion of the houses in each grid should be assigned to the municipality. They are the proportion of the number of buildings in each grid and municipality intersection with respect to those in each grid. Similarly, for a grid wholly contained within a municipality, all the houses damaged in the grid would be assigned to the municipality.

6. Line 385: "...experiencing greater than 10% severely damaged houses would receive relief" What does "10% severely damaged"? Is this the same as the abovementioned "10% damaged"?

A: We apologize for the vagueness of the language here. Indeed it is the same measure of being damaged. This refers to 10% of all the houses falling into the NDRRMC category of "completely damaged" (so not inhabitable based on visible inspection). We will remove the "severely" from the next version of the paper.

7. The manuscript has some grammar errors and incomplete sentences somewhere. For example, in Lines 377-378 "The G-Naive model never predicts the damage will be over 10%, as most of the data skews to minor damage." ; Line 382 "This results in a 6.9% Overall, the F1 measure of the proposed model is the best at 0.65". I suggest that the authors thoroughly revise the language.

A: Thank you for pointing out this awkward language. We will modify it in the next version of the paper. We will furthermore again revise the whole text of the manuscript for grammar errors and incomplete sentences.

8. Discussion & Conclusions: It would be beneficial if the authors could discuss why your grid-based model with fewer features outperforms the original model, particularly in the high-damage categories. Is this improvement due to the use of higher-resolution data capturing spatial details? Could the inclusion of socio-demographic features account for the improved performance of your model?

A: Thanks for the suggestion. We already hinted as this in the discussion when we state that

"It should be noted that the grid-cell model can distribute some of its predictions to municipalities outside the municipalities of the target training data, with two consequences. It allows the uncovering of areas that might have been overlooked in the data collection (e.g., areas with mild damage that have not been reported but are helpful to train the model). However, it also might allocate damage to areas of minor importance (e.g., neighboring villages with very few houses)."

We will extend these sentences in the manuscript stating that "Furthermore, this redistribution of data to the grid-level leads to a more fine-grained and uniform spatial data distribution. This augmentation of datapoints (as can be observed in Figure 2) quite possibly has a positive influence. Additionally, also the inclusion of socio-demographic features like the RWI index and terrain type has certainly helped in improving model performance."

**CC1, Chu-En Hsu, 30 Oct 2023**

The prior typhoon impact forecast model's performance has been improved by the present work. While the authors have done an excellent job of investigating historical severe typhoon events affecting the Philippines, I would like to draw attention to the relevant analyses and discussions on storm surges and wave runup during three historical hurricanes (Matthew 2016, Dorian 2019, and Isaias 2020) studied by Hsu et al. (2023; https://doi.org/10.5194/nhess-2023-49) using a numerical modeling system (i.e., COAWST; Warner et al., 2010). Along the South Atlantic Bight, the relative contributions of storm surge and wave runup were analyzed. Hsu et al. (2023) also studied the connection between storm characteristics (e.g., storm translation speed and wind speed) and variations in the water level components.

A: The current impact-based forecasting model uses a storm surge susceptibility map, which is -we fully understand- a static map and hence a simplification as it does not cover the dynamics. We acknowledge the importance of and are aware of the developments in the area of storm surge modeling. We are in a project around Destination Earth to assess the use of GTSM, see https://stories.ecmwf.int/destine-digital-twins-to-anticipate-the-devastating-effects-of-flooding-in-coastal-areas/index.html. Future developments of the IBF Model might include hybrid modeling where the machine learning model exchanges the static storm surge map with a hydro-meteorological dynamic model. We speak about this briefly in the paper: "For example, for storm surge, dynamic models, such as the Global Tide and Surge Model, are being developed (Bloemendaal et al., 2019). Incorporating these dynamic models into an ML model might improve the secondary hazard features used in grid- and municipality-based models. The new technologies being developed as part of Digital Earth (Annoni et al., 2023) can play a role here." We would be happy to add further citations to the final version of the paper.

**RC4, Nadia Bloemendaal, 03 Nov 2023**

Please use abbreviations consistently throughout the manuscript (and make sure they are properly introduced). For example, the manuscript text jumps back and forth between TC and tropical cyclone; this should be homogenized to TC.

A: Thanks for pointing out the inconsistencies in our use of abbreviations. In the subsequent version of the paper we will proof-read it and standardize their use.

Please also make sure that the term "disaster" is correctly used at the respective instances, or whether the term "natural hazard" is more appropriate (a natural hazard becomes a disaster when people are involved).

A: We will examine the use of "disaster" language and change it when it is not appropriate.

I noticed that the authors use IBTrACS as input dataset. Are the authors aware that IBTrACS reports wind speeds in the Western North Pacific as 10-min average sustained wind speeds? Table 2 mentions wind speeds as 1-min sustained values, however I cannot find any information on possible conversion factors being applied (I might have missed this). In this light, I also wanted to point out that wind speeds on the Saffir-Simpson scale (indeed) are given as 1-min values, so if the IBTrACS values weren't converted, the Saffir-Simpson categories also need conversion (see Bloemendaal et al 2020, https://doi.org/10.1038/s41597-020-00720-x for the Saffir-Simpson values in 10-min sustained wind speeds)

A: We use the statistics provided by the US agencies (NOAA & JTWC), which report 1-min sustained wind values, so no conversion is done. https://www.ncei.noaa.gov/sites/default/files/2021-07/IBTrACS_version4_Technical_Details.pdf The rest of the agencies report their data in 10-minute sustained values, but we do not use these.

One other comment I have about the use of the Saffir-Simpson scale is that it can be confusing to use the Saffir-Simpson categories in relation to typhoons. While I understand that, for the global audience of this article, it can be more interpretable to use the Saffir-Simpson scale, I also find it confusing to read a sentence like "We choose Melor, a powerful typhoon of category 4 that struck the Philippines in December 2015." (line 392-393). This is primarily because a typhoon of Category 4 could locally be perceived as a "very strong typhoon" or a "violent" typhoon (depending on how one would interpret the category). Perhaps it's an idea to clearly communicate that the Categories are given as categories on the Saffir-Simpson scale at all relevant instances.

A: Indeed, the typhoon is classified as a very strong typhoon by Japan Meteorological Agency and category 4-equivalent typhoon by JTWC. We are using the Saffir-Simpson scale here, despite the geography of the typhoons studied, and we will clarify this in the text. We will also clarify the use of Saffir-Simpson scale in the caption of Figure 3.